# Comparison of total ozone measurements in Melbourne, Australia, performed with a low-cost micro spectrometer and a Brewer MK-III

Kåre Edvardsen[1], Matt Tully[2], Steve Rhodes[2]

[1]Department of Automation and Process Engineering, UiT, The Arctic University of Norway, Tromsø, 9018, Norway
[2]Bureau of Meteorology, 700 Collins Street, Docklands, Victoria 3008, Australia

*Correspondence to*: Kåre Edvardsen (kare.edvardsen@uit.no)

**Abstract.** A new design for a simple and very low-cost instrument for measuring total column ozone is described and its performance evaluated by comparison to a co-located MK III Brewer spectrophotometer for approximately six months. The ozone retrieval is based on the "Global Irradiance" method used to derive total column ozone from the Norwegian GUV radiometer network. While the total cost of components of the instrument was less than EUR 3000, total ozone values were found to agree with Direct Sun Brewer values with a standard deviation of 1.8%, comparable to the agreement between DS and ZS Brewer data. The most significant limitations of the retrieval method were found to be the dependence on temperature and cloud-amount. As well as being low-cost, the instrument appears robust and easy to operate. As the instrument covers fully the UV-A/B spectrum it is fully possible to further develop the analyses software to include other data products related to UV-radiation.

## 1 Introduction

Measurement of the atmospheric ozone layer has been of scientific interest for more than 100 years, with G. M. B. Dobson being acknowledged as the first scientist to be able to routinely measure total ozone column (TOC) with the famous "Dobson spectrophotometer", developed during the 1920s (Dobson & Harrison, 1926) and becoming available in its current form from the early 1930s. It was not until the 1970s, however, that concerns first arose about a potential major depletion of the earth's ozone layer as a consequence of human activities (Crutzen, 1979; Molina & Rowland, 1974; Rowland & Molina, 1975). The Brewer spectrophotometer was designed during the early 1980s for automated, accurate measurements of TOC (Kerr et al., 1981), and over the coming years, it became one of the most reliable sources for monitoring the TOC through a worldwide network of more than 200 instruments. Most of these instruments are included in the "Global Atmosphere Watch Programme" (GAW, https://public.wmo.int/en/programmes/global-atmosphere-watch-programme administrated by the World Meteorological Organization WMO, https://public.wmo.int/)(Tully, 2020).

In recent years, new technology has enabled the development of much cheaper instruments for measurements in many fields of atmospheric science (Serrano et al., 2022; Kanellis, 2019; Turner et al., 2020). Such "low-cost sensors" do not usually have the same performance as more traditional instrumentation but offer numerous advantages including the possibility of

much denser spatial coverage. To some extent, newer technology has also started to become available for measurements of total column ozone (Herman et al., 2015; Zuber et al., 2021; Gkertsi et al., 2018; Michalsky and Mcconville, 2024; Geddes et al., 2024)  however, all of these instruments still cost a significant fraction of the price of a new Brewer to purchase and so cannot be considered "low cost", despite repeated calls

(https://ozone.unep.org/system/files/documents/ORM11_Recommendations.pdf, Capacity Building section). As well as the

initial purchase price, existing TOC instruments typically need very frequent inspection and maintenance and require calibration at regular intervals. Calibrations against the recognised standard are difficult logistically as well as being very costly and beyond the means of most countries to carry out without external assistance.

The main objective of this work is to describe and investigate the performance of a novel design for a measuring instrument (which we call the STS being based on the Ocean Optics STS-UV micro-spectrometer) in order to introduce a genuinely low

cost and user-friendly alternative to the established instruments. Research which requires costly initial investment and high ongoing expense can serve as a barrier to developing countries participating in the relevant international research communities. The STS instrument has a total price in the order of less than 1.8 % of the cost of a Kipp & Zonen, Brewer MK-III instrument, and is significantly easier and less costly to operate. This opens a much greater opportunity for developing countries to be able to invest, not only in instrumentation, but also the required scientific and technical training in

order to achieve scientific progress in environmental research.

In this study, the STS instrument's performance over time is investigated by comparing the STS measurements with DS measurements from a co-located MK III Brewer, with a double monochromator which has insignificant stray light impact, and may be regarded as a reference for ground-based TOC measurements.

## 1.1 GI ozone retrieval

The Brewer is designed to measure both the direct solar radiation (DS) and scattered solar light straight from the zenith (ZS), by alternately directing the optical detector towards the sun and the zenith of the sky. Intensities at five distinct wavelengths at around 306 nm, 310 nm, 313 nm, 317 nm and 320 nm (Brewer WLs), are recorded for further analyses. The principle of calculating the TOC from the DS measurement is well known and it relies on the absorption of UV-radiation by ozone through the atmosphere, according to the Beer-Lambert law. Corrections are made to the measurements, mainly to allow for

absorption and scattering by $SO_2$ and atmospheric constituents, respectively. The DS-method has proven to be highly accurate in studies already 20 years ago (Fioletov et al., 2005), but there has been done some work lately to further improve the DS-measurement (Savastiouk et al., 2023). Accurate DS measurements rely on a clear path between the instrument and the sun with only negligible UV absorbing constituents in the atmosphere, other than ozone. However, the presence of even thin clouds is the main reason for the rejection of the DS measurement as the clouds heavily increases the uncertainty in the

DS data, and therefore an additional Zenith Sky (ZS) measurement is performed and the intensities at the Brewer WLs are recorded regardless of the sky conditions. During the campaign the total number of DS measurements was almost 7400 and 34.5 % of them were rejected due to higher standard deviation than 2.5 DU which is the rejection criterion. The idea behind

the alternating DS and ZS measurements originally came from G. M. B. Dobson, and assumes a consistent statistical relation between accurate DS and following ZS measurements. Once the relation is established, the ZS measurement can be used in absence of DS measurements, in order to have more continuity in the ozone record. This is particularly of interest at sites with cloudier conditions, such as high latitudes or generally in areas with more unstable weather conditions.

In addition to DS and ZS, a less-commonly used third mode of operation of the Brewer is also available known as global irradiance "GI". For GI measurements the sum of the direct and diffuse radiation falling on a flat horizontal surface is measured through the UV dome of the Brewer instead of the flat quartz window used for DS and ZS measurements. The principle behind this method (GI TOC) is well described in (Dahlback, 1996; Stamnes et al., 1991; Høiskar et al., 2003), and, unlike DS and ZS, requires comparing measurements with modelled data to derive an ozone value.

The intensities of GI measurements will be influenced by both clouds as well as ozone which will affect the TOC retrieval. In order to reduce this kind of cloud effect, the fit between the measurement and model data is performed with ratios between two intensities at wavelengths where one is significantly more absorbed by ozone than the other, and at the same time both intensities must be almost equally affected by clouds. In this way the cloud effect will be significantly reduced, although the retrieval will still only be valid for smaller cloud effect situations (Stamnes et al., 1991). This method also has another advantage, which is that there is no need for an absolute irradiance calibration of the measurements as a measured ratio is used (the units are cancelled), rather than using measurements of absolute irradiances. Norwegian Institute for Air Research (NILU) developed a limited series of UV-radiometers sold worldwide, the NILU-UV (Høiskar et al., 2003), which has been used in several studies (Kazantzidis et al., 2009; Fan et al., 2015; Zhao Di, 2018; Lakkala et al., 2019; Sztipanov et al., 2020; Lakkala et al., 2005) over a large range of latitudes and altitudes, suggesting a method that works well under clear or moderately cloudy sky conditions. The method for TOC retrieval is also applied to the Norwegian network of UV radiometers from Biospherical Instruments Inc. for the survey of solar UV radiation, with a data record going back more than 30 years (Svendby et al., 2021). While some studies suggest less effects by clouds (Dahlback, 1996) the impact from heavy cloud situations in other studies (Mayer et al., 1998) shows overestimation of the TOC when compared to DS measurements.. It will be shown in this study that increased cloudiness in general gives an overestimation of the TOC based on GI measurements performed by the low-cost, Ocean Optics STS-UV micro spectrometer (STS).

It should also be mentioned that, in addition to the DS, ZS and GI modes, it is possible to derive total ozone from the Brewer's spectral UV irradiance using a statistical relationship (Fioletov et al., 1997).

## 2 Material and methods

### 2.1 Instruments

The Bureau of Meteorology, Australia (BOM), operates several Brewer MK-III instruments for measurements of TOC in both DS and ZS mode, to complement the long-standing Dobson record in Australia (Tully et al., 2015). The comparison took place at the Broadmeadows Training Centre, 144.946 °E, 37.691 °S, around 25 km north of Melbourne city centre.

Broadmeadows is a registered site at the World Ozone and Ultraviolet Radiation Data Centre (WOUDC), Network for the Detection of Atmospheric Composition Change (NDACC), Pandonia Global Network (PGN) and Eubrewnet with several co-located ozone measuring instruments. The Bureau of Meteorology Brewers are regularly calibrated by International Ozone Services to ensure traceability to the World Calibration Centre operated by Environment and Climate Change Canada. The last DS calibration prior to the campaign was May 22, 2018, and the ZS coefficients in use were recalculated

March 21, 2019.

The Ocean Optics STS-UV is a small size spectrometer, only 4.0 x 4.2 x 2.4 cm$^3$, covering the range of ~190 – 650 nm with a triangular slit function of ~1.5 nm FWHM (Ocean Optics lab test). The complete instrument function was not provided by Ocean Optics, but from comparing measurements and model simulations, it was estimated to have a triangular form with a FWHM of about 3 nm in the 310-320 nm region. when using a 25 μm slit. The optical configuration is an asymmetric

crossed Czerny-Turner model with a focal length of 28 mm. The detector is a Panavision, ELIS-1024, linear array detector, which consists of 1024 high performance, low dark current photodiodes. The diode signals are digitized with a 14-bit ADC, giving a dynamic range of 0-16383, where the dark signal varies between approximately 1500-5000, depending on temperature. It is run by a Raspberry Pi Model B microcomputer via an USB connection supporting both power and communication with the STS. Nominal power consumption for the system (STS-kit) is about 500 mA at 5.0 V, but may

reach up to 1.2 A during start up. It is recommended to use at least a 2 A power supply to ensure that the STS-kit runs within a reasonable safety margin with respect to power consumption. This will put less stress on the power supply electronics and minimizing the risk of a power failure.

The STS and the microcomputer fit easily in a commercially available IP67-rated, UV resistant enclosure box (110 x 180 x 70 mm), with an externally mounted cosine diffuser for irradiance measurements. The diffuser has a 20 mm diameter flat

surface, made of 0.2 mm thick PTFE folio. The diffusing surface is placed approx. 30 mm above the entrance slit, ensuring a sufficient covering of the spectrometer's field of view (FOV) of 25°. Memory usage with a measurement frequency of one sample per minute is around 0.9 GB per week, allowing the instrument to run continuously for more than one year with a 64 GB memory card. The total cost of components was less than EUR 3000.

## 2.2 Data record

Data was recorded from both the Brewer and the STS during January 1 – June 18, 2019, at the Broadmeadows site. The Brewer was set up to perform DS TOC measurements around 15 times per hour for SZA less than approximately 63°, where one single approved measurement consists of an average of five consecutive measurements for which the standard deviation (STD) must be less than or equal to 2.5 DU, ensuring the data to be less affected by other constituents (typically clouds and heavy aerosols) than ozone. The Brewer was running continuously and recorded about 80 DS and 14 ZS measurements on

clear sky days in early January, and less frequently on cloudy days depending on the cloud situation. By the middle of June, the number of DS observations were reduced to around 25 on a clear day, as the period of SZA less than 63° became less than three hours a day. Above 63° the Brewer was doing umkehr instead of DS and ZS measurements. During the whole of

January, the STS detector saturated for SZA less than around 45° because of a too long integration time of 10 seconds. (The integration time was shortened for the remainder of the campaign to prevent this problem recurring). Only data recorded between SZA of 45° and 63° are therefore considered for January. Some data was lost by the STS because of a thunderstorm incident in the evening on February 6[th], causing an instrument power failure. The instrument was back running again around noon February 11[th]. Apart from this power failure, the STS was running continuously and recorded one minute mean raw data regardless of the SZA. Processed TOC data recorded with SZAs less than 63° are used for the analyses.

## 2.3 Total ozone column retrieval

The original Brewer DS-TOC retrieval method is well-described in (Kerr et al., 1985) and is based on absorption of UV-radiation through the atmosphere according to the Beer-Lambert law, assuming the sun as a point source with parallel incoming rays. The Brewer ZS TOC retrieval relies purely on an assumed statistical relation between DS and ZS observations through a compound quadratic function (model) of DS-TOC and the ozone slant path (air mass factor) described as:

$$F - F_0 = (a\mu^2 + b\mu + c)X^2 + (d\mu^2 + e\mu + f)X + (g\mu^2 + h\mu + k) \quad (1)$$

where $a, b, c, \dots, k$ are the equations coefficients, $X$ is the DS-TOC observation value, $\mu$ is the air mass factor, and $F - F_0$ is the difference of the logarithmic sum of measured ZS intensities $F$, and the expected extra-terrestrial value $F_0$ of the same logarithmic sum of intensities. The coefficients are statistically derived through a large set of sequential DS and ZS measurements over a large range of $X$ and $\mu$ values, and cloud conditions (usually in the range around 1000 measurements or more) to ensure all coefficients to be statistically significant in the model. Once the coefficients are derived, $X$ may be solved as the unknown in a standard quadratic equation.

TOC retrieval from GI measurements is based on the multi-stream discrete ordinates radiative transfer equation solver DISORT, first described in (Stamnes et al., 1988; Stamnes et al., 1991), assuming a plane parallel atmosphere and later refined to include pseudo-spherical corrections (SDISORT) as described in (Dahlback and Stamnes, 1991). The SDISORT has later been included in a software tool called libRadtran (http://www.libradtran.org/doku.php?id=start) and the version 2.0.2 of December 2017 (Emde et al., 2016) was used for the calculations in this work. The software package is a library of C and Fortran functions and programs for calculation of solar and thermal radiation in the Earth's atmosphere.

The libRadtran tool takes input parameters describing the optical properties of the atmosphere and the earth's surface and calculates the global irradiance at a given location (latitude, longitude and altitude). In order to compare GI measurements with model calculations libRadtran offers a tool that weights the calculations over a certain wavelength range with a specified bandpass function in accordance with the instrument used for the measurements. In this way it is possible to compare measurements and simulated model results as the model mainly represents the characteristics of the instrument through the applied band pass function to the simulations.

The measured irradiance data is in general calculated by fitting the measured ratio between the irradiances at 321.9 nm and

312.2 nm, $R_{me} = \frac{I_{321.9\,nm}}{I_{312.2\,nm}}$, with a corresponding modelled ratio $R_{mod}$. As the GI measurements are not absolutely calibrated, the $R_{me}$ need to be scaled with a constant $k$, so that $R_{mod} = k \cdot R_{me}$ for a known TOC value. In order to minimize errors in deriving $k$, the TOC value should be taken from a Brewer DS measurement retrieved closely in time with the according $R_{me}$. Once $k$ is estimated, the relation between $R_{mod}$ and $R_{me}$ is assumed to only be dependent on the change in TOC (this assumption has its limitations and is handled in Sect. 3). For simple and fast STS TOC retrieval a lookup table (LUT) is

calculated covering the range of SZA's and TOC for the site of interest. The $R_{mod}$ in this work was calculated for TOC values ranging from 100-600 DU with 20 DU steps at SZA values ranging from $0° - 90°$ with 1° steps. As both the $R_{mod} = k \cdot R_{me}$ and SZA is known, the TOC is derived through linear interpolation between the LUT values. Figure 1 shows the resulting surface of the $R_{mod}$ with the corresponding SZA and TOC.

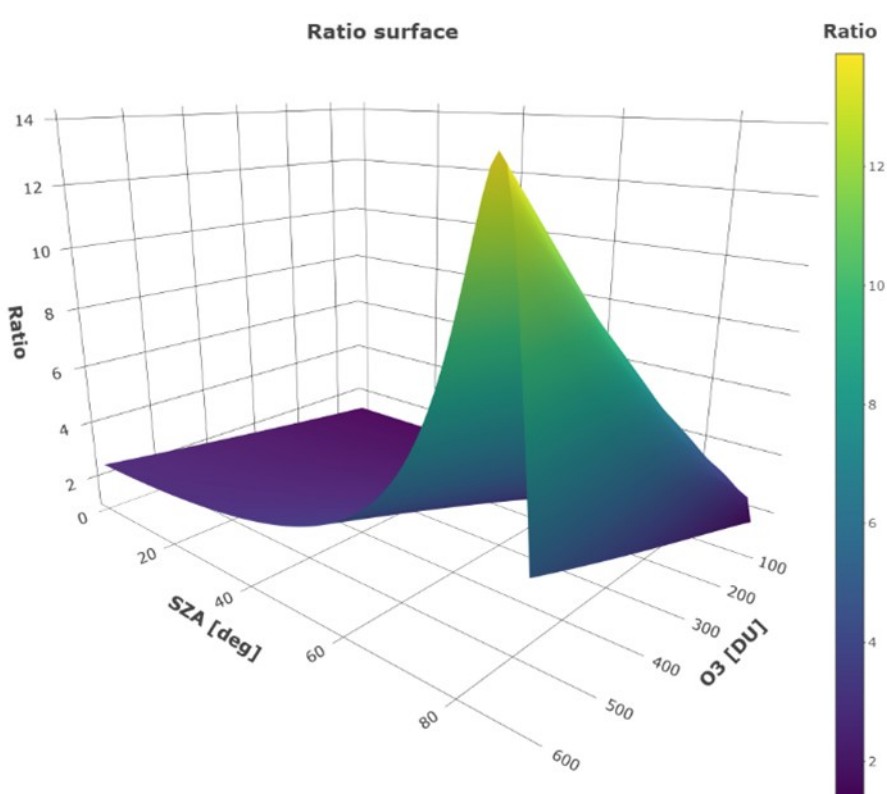


**Figure 1: The resulting surface from the linearly interpolated LUT values.**

The model input atmosphere for the calculations was derived by the Air Force Geophysics Laboratory (Anderson et al., 1986) corresponding to a season dependent, mid-latitude, clear sky situation (AFGL). That implies no additional model input

with respect to atmospheric constituents are used in the calculations. Other fixed input parameters to the model were from the extra-terrestrial irradiance spectrum from the Atlas-3 mission (Thuillier et al., 1998), the radiative transfer solver was SDISORT, and the ozone cross section was "Bass and Paur" (Bass and Paur, 1985), to match the operational Brewer retrieval. In addition, libRadtran sets default values for other input variables in order to reflect normal atmospheric conditions. See (Mayer and Kylling, 2005) for details.

## 2.4 Influence of clouds and ozone profile


As clouds will enhance the absorption of UV-radiation due to multiple scattering (Mayer et al., 1998), an overestimation of the TOC will result. In order to exclude such overestimated TOC data a simple cloud detection routine (CDR) was applied in the calculations. The CDR calculates the ratio between a measured and a modelled GI at a specific wavelength where the absorption by ozone is negligible (340 nm was used in this case), where the modelled GI represent a clear sky situation. The

raw data at 340 nm was calibrated against a modelled clear sky $GI_{340nm}$, where $GI_{340nm} = 0.56 \cdot GI_{raw\ 340nm}$. If the calibrated ratio is close to unity, a clear sky condition is assumed, and a ratio of 0.9 indicates a reduction in irradiance of 10 % due to a cloudy condition. Whenever the ratio drops below the predefined threshold the TOC calculation is flagged as a cloudy condition measurement. The setting of the threshold value is not based on objective calculations but is a user selection depending on requirements, in that the lower threshold (more cloudy condition accepted) the higher risk of

overestimation and variation during the measurement period. For this paper a threshold value of 0.9 was used, as the effect of a lower threshold than this will include too many scenarios with cloudy conditions, which again will lead to overestimation and loss of TOC data.

As the ozone profile is well known to change both by season and the meteorological situation it may influence the surface UV radiation differently even if the TOC is constant. This is also an error source in the TOC derivation from GI-

measurements as the assumed, constant ozone profile in the model may differ significantly from the actual profile (Lapeta et al., 2000). In this study the ozone profile as presented in (Anderson et al., 1986) is used under the assumption that uncertainties related to the ozone profile are not significant for SZA's less than 63°.

## 2.5 Influence of temperature

The Broadmeadows site in Melbourne experiences a wide range of temperatures from just above 0°C wintertime up to 45°C,

or even higher on some occasions during summer. Unfortunately, the manufacturer (former "Ocean Optics") of the micro spectrometer has no information on temperature dependence with respect to either sensitivity or wavelength. It is commonly known that both electronics and mechanics in electro-optical systems are affected by temperature changes, like electrical noise, and thermal expansion. This will likely introduce some uncertainty in the measurements which is partly corrected for in this work. As the micro spectrometer did not provide detector temperature it is not possible to perform an accurate

correlation study between the temperature and the GI-TOC results. However, by using the ambient temperature $T_a$ measured by the Brewer instrument it was possible to perform a linear regression based on the GI-TOC measurements which were

clearly increasing with increasing temperatures when compared with the DS measurements. The applied correction factor $C_t$ to the GI-TOC was calculated based on the simple linear regression

$$C_t = 5.0 \cdot 10^{-3} \cdot T_a + 0.90 \quad (2)$$

where $T_a$ is in °C. At a temperature of $T_a = 20\,°C$, $C_t = 1.0$ (no correction) and $T_a = 30\,°C$, $C_t = 1.05$ (5% correction). Ambient temperature at the site during the measurement period varied form a daytime maximum of 45.8 °C on Jan 25[th] to a daytime minimum of 10.0 °C on May 30[th], which is within both ends of the recommended operating temperature range for a running STS (0 – 50 °C). After applying the temperature correction to the data, the TOC on the warmest day was measured to 268.5 DU for the Brewer and 275.8 DU for the STS (+2.7 %). On the second warmest day with 44.3 °C on Jan 4[th], the

Brewer measured 269.1 DU and the STS measured 268.9 (-0.1 %). On the coldest day the Brewer measured 304.5 DU and the STS measured 306.5 DU (+0.7 %). This indicates that the temperature has an influence on the STS instrument and that the correction routine improves the results, as shown in figure 2 below. Panel (a) shows the ratio between the STS TOC and Brewer DS measurements for clear sky conditions over almost the entire temperature range during the campaign. The linear fit is given in equation (2) and is used for temperature correction of all STS TOC measurements. Panel (b) shows the ratio

between the uncorrected daily mean STS TOC and Brewer DS measurements, indicating a general overestimation in the STS TOC measurements. After applying the temperature correction (c) the slope is close to unity with a small offset of only 2.1 DU.

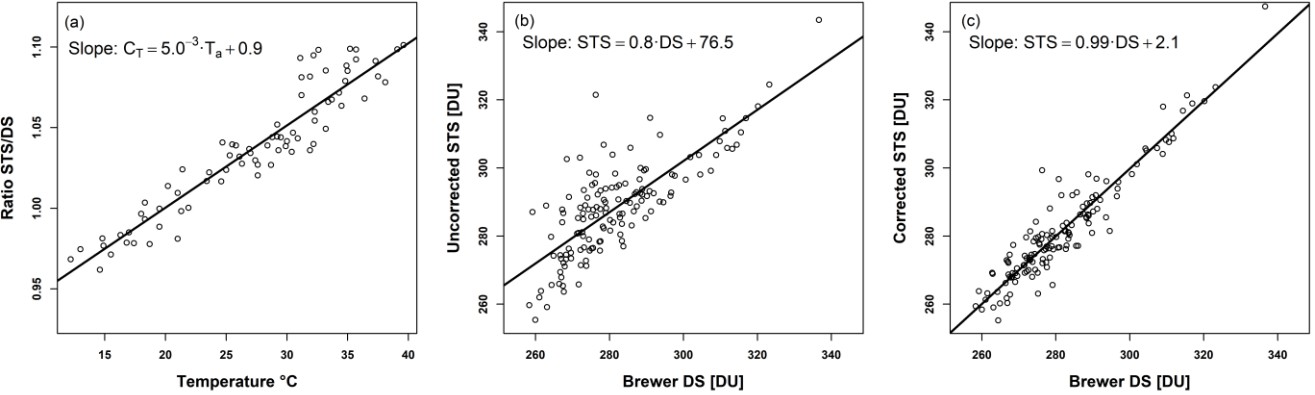

Figure 2: (a) Ratio of STS derived TOC to Brewer DS as a function of Brewer temperature, showing an

approximately linear increase in TOC values with increasing temperature. (b) Scatter plot of STS TOC values against Brewer DS without temperature correction.  (c) Scatter plot after applying the temperature correction shown in panel (a). The regression slope is close to unity with a small offset of 2.1 DU.

## 3 Results and discussion

This work mainly focuses on daily mean data, and a selection of high temporal resolution data from days with both clear and overcast skies. The daily means are calculated independently for both instruments, and always with SZA's less than ~63° (see section 2.2). The number of approved daily means was 148, 161, and 135 for the Brewer DS, Brewer ZS and STS clear sky, respectively. The daily mean reference DS data showed that the minimum and maximum TOC reached 258.3 DU and 336.6 DU on May 12 and March 30, respectively. The average DS measurement was 281.7 DU during the measurement period, where the ZS and STS average results are very similar (see Table 1 below).

| Instrument | Nr. of daily means | Min TOC [DU][b] | Max TOC [DU][b] | Mean TOC [DU] | Standard deviation [DU] |
|---|---|---|---|---|---|
| Brewer DS (ref) | 148 | 258.3 | 336.6 | 281.7 | 14.5 (5.1%) |
| Brewer ZS | 161 | 266.8 | 343.1 | 287.6 | 20.6 (7.2%) |
| STS, clear [a] | 135 | 259.4 | 347.4 | 281.1 | 15.6 (5.5%) |
| STS, clear and cloudy | 163 | 263.9 | 337.5 | 293.2 | 19.2 (6.5%) |

[a] The STS did not measure Feb. 7 – 10 due to a power failure caused by a thunderstorm. All other missing days for any of the observations were related to heavy overcast skies.

[b] Min and max TOC measurements are compared at the same dates as the Brewer

**Table 1: Main results overview.**

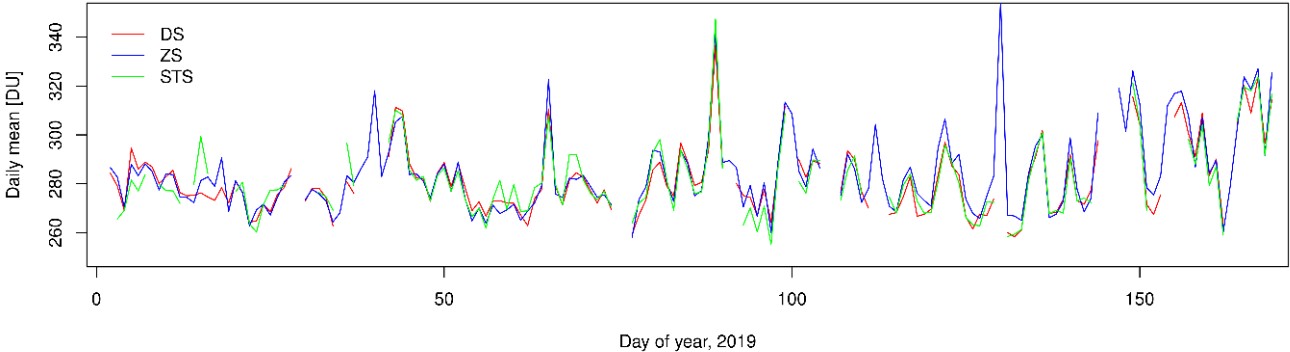

**Figure 3: Series of daily mean. Both the ZS and STS TOC measurements correlate well with the DS reference measurements. It is worth noticing that on days with missing reference data the ZS measurements are usually higher than the mean value from the DS measured before and after the days of missing values. This is clearly seen around day 40, 112, and 130.**

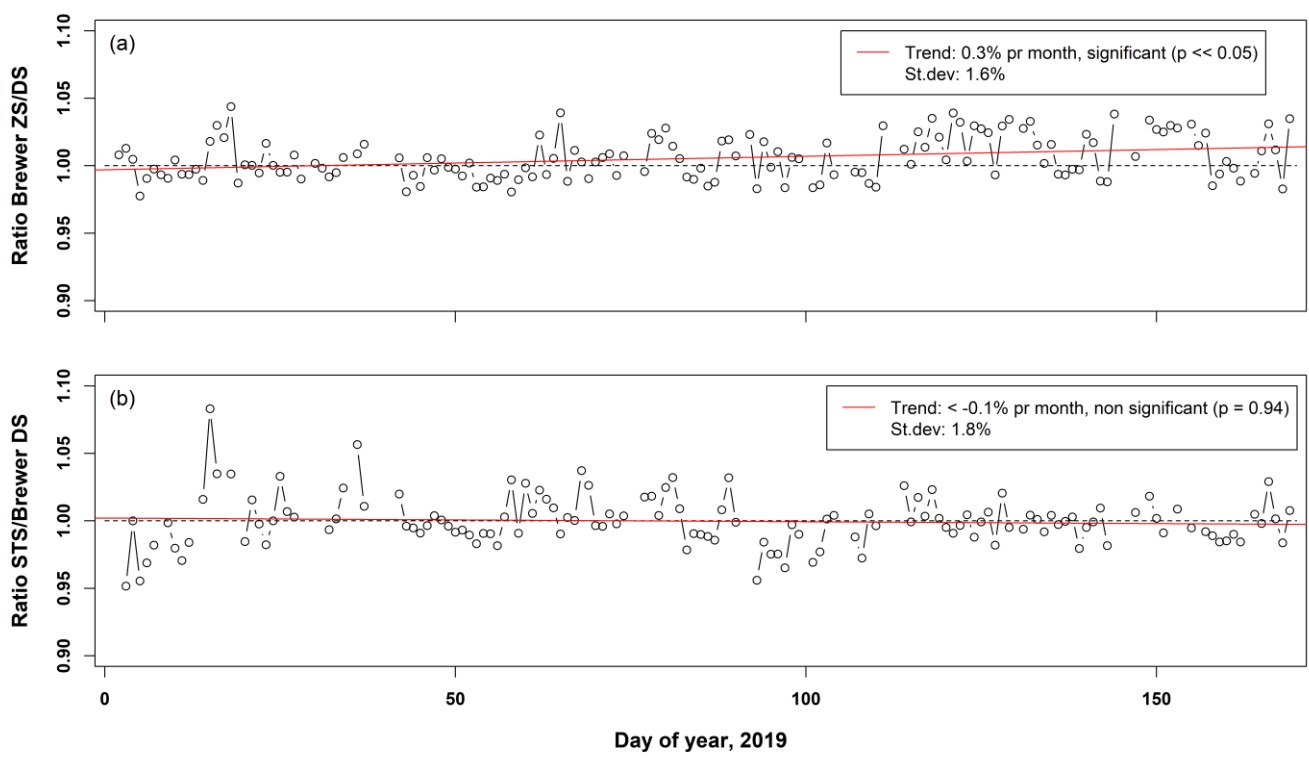

**Figure 4: Compared daily mean ratios between the Brewer ZS and DS measurements (a) and the STS TOC and Brewer DS measurements (b).**

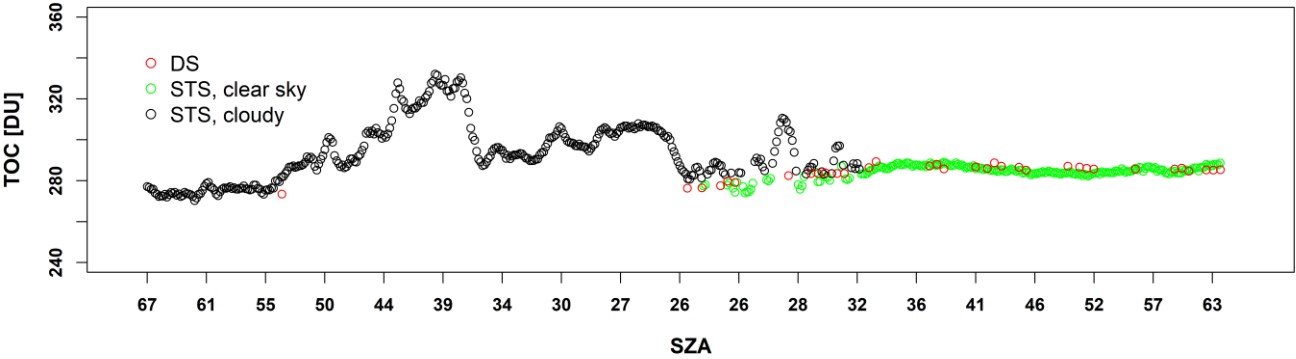

**Figure 5: This shows a typical plot with cloudy conditions before noon where the STS TOC measurements are increased and highly varying. During afternoon the variation is reduced, resulting in better agreement with the Brewer DS measurements.**

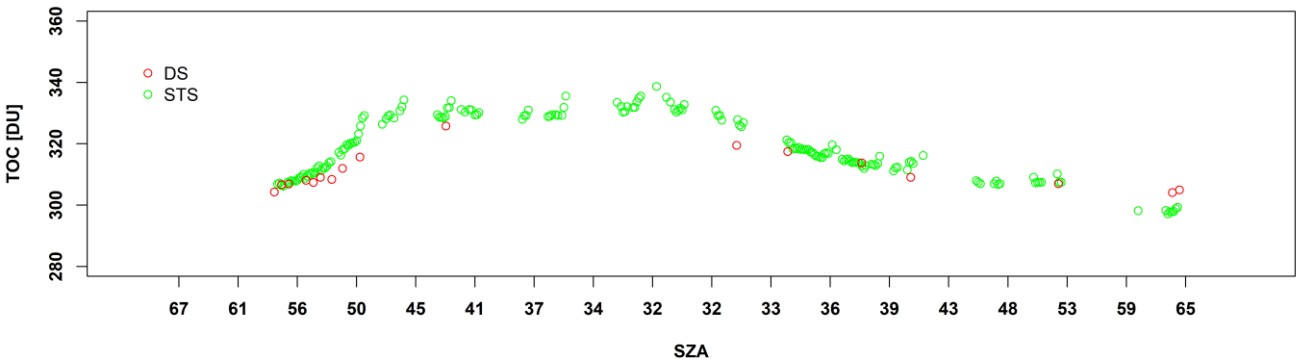

**Figure 6: Variation in the TOC for the day with the second highest difference between highest and lowest value (21.7 DU). The relatively few DS measurements suggest that they were affected by clouds.**


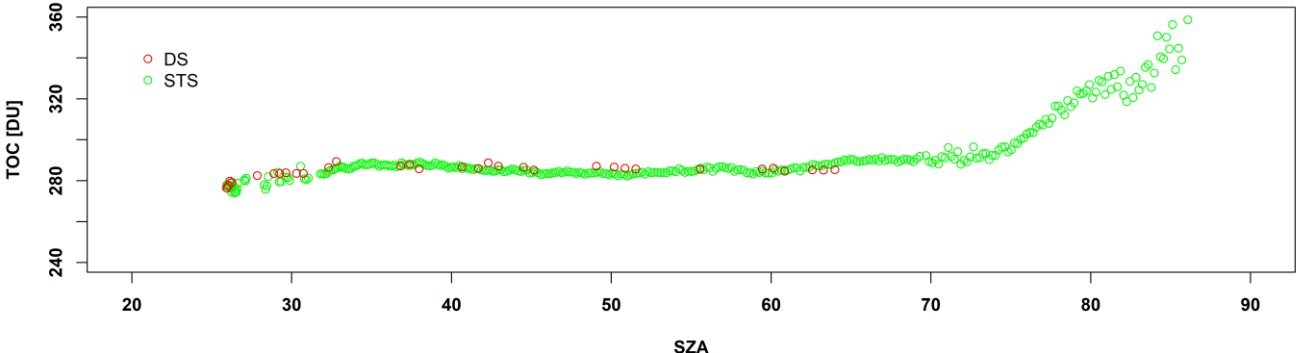

**Figure 7: When the SZA approaches 70°, the detection level is almost reached, and the model predicts this as a false increase in TOC as the 312.2 nm intensity disappears earlier than expected.**

### 3.1 Data assessment

Missing DS measurements (see figure 3 around day 40, 112, and 130) are most likely a result of too cloudy conditions, but the ZS measurement may still be recorded within the accepted limit of STD of 2.5 DU. The possible over estimation of the ZS TOC in this situation could be a result of an increased absorption of UV-radiation due to multiple scattering in the clouds (Mayer et al., 1998) rather than absorption from increased atmospheric ozone. This assumption is indicated in figure 5 where the black circles are STS measurements done under cloudy conditions at day 49, showing unrealistic variation and increase
in the TOC. As the sky becomes clearer during the afternoon (green circles), the measurements are less affected by clouds

and in more agreement with the DS data. The same effect is not investigated for the Brewer ZS measurements in Melbourne, but it is clearly a subject for further work. It should be noted that the STS measures global irradiance and may be more sensitive to multiple scattering, as the whole sky is regarded, than the Brewer ZS measurements where the incoming light has originated from only a small solid angle of zenith sky. When regarding all the clear and cloudy sky STS measurements for SZA's < 63°, the campaign mean value (table 1) increases by 4.3 %, from 281.1 DU to 293.2 DU, and the standard deviation increases from 15.6 DU to 19.2 DU, as the daily variation is much higher.

During the campaign the TOC did not vary a lot while the Brewer performed DS measurements. More than 95% of the days had a difference between the highest and lowest value less than 10 DU, and only three days with a difference higher than 20 DU. The day with the highest variation in TOC was day number 5, where the lowest and highest measurement was 282.6 DU and 321.1 DU, respectively (38.5 DU difference). Unfortunately, almost all the STS measurements were rejected this day due to both saturation problems and too overcast conditions. Only two data points around the lower value were recorded. The day with the second highest variation in TOC was day number 65, where the lowest and highest measurement was 304.1 DU and 325.8 DU (21.7 DU difference). Figure 6 shows how the TOC increases relatively rapidly towards noon, before slowly decreasing during the afternoon.

Assessing time series between instruments often includes studies of the ratio between the measurements for analysis of trends and standard deviation (STD). In panel (a) of figure 4 the Brewer ZS/DS is shown and there is only a slight, but statistically significant trend of + 0.3 % per month, with an overall STD of 1.6 %. The non-zero trend between summer and winter suggests a small seasonal dependence in the statistical relation between ZS and DS presumably due to the distribution of clouds and ozone.

In panel (b) the STS/DS ratio is shown, and no significant trend was found. A slightly higher overall STD of 1.8 % was found. The increase in the STD is mainly connected to the larger variations in the ratio during the first 30 days of the series when the measurements are only considered for SZAs higher than 45° due to detector saturation in the STS instrument. This was also by far the warmest period during the campaign, and the temperature correction may fail due to much higher internal instrument temperature than the ambient temperature used for the correction. However, this is only a possibility and the real reason for the larger variation remains unexplained. In all, the accuracy of the STS when used for daily mean TOC is in the same order as for the Brewer ZS measurements. Similar results are found for other Brewer instruments in the GAW network (the site at Environment Canada, Toronto, Ontario), with STDs of 1.7 % between DS and ZS values for two of the instruments (Fioletov et al., 2011).

When it comes to the STS's applicability for SZA's higher than 60°, it is clear that the measurements are less reliable. For the STS, the main reason is the relatively low detector sensitivity combined with high dark signal, resulting in a very low signal to dark ratio (low dynamic range). For SZA's ≈ 70° around 85% of the measured intensity at 312.2 nm, is the dark signal, and when the SZA approaches 85°, the detection level is almost reached (found by inspection of the raw data). The model predicts this as a false increase in TOC as the 312.2 nm intensity is almost comparable to the dark level. The effect can clearly be seen in figure 7. From figure 8 it is clear that the STS performance is good for SZA's less than 70°, assuming

a stable TOC the rest of this day (data from the same day is presented in figure 5 and 7). When looking at the STS/DS ratio in detail, the difference never exceeded 1.7 % and the standard deviation is only 0.8 %. Comparing measurements this way requires the measurements to be recorded simultaneously, and in this case each ratio is from data recorded within one minute time difference.

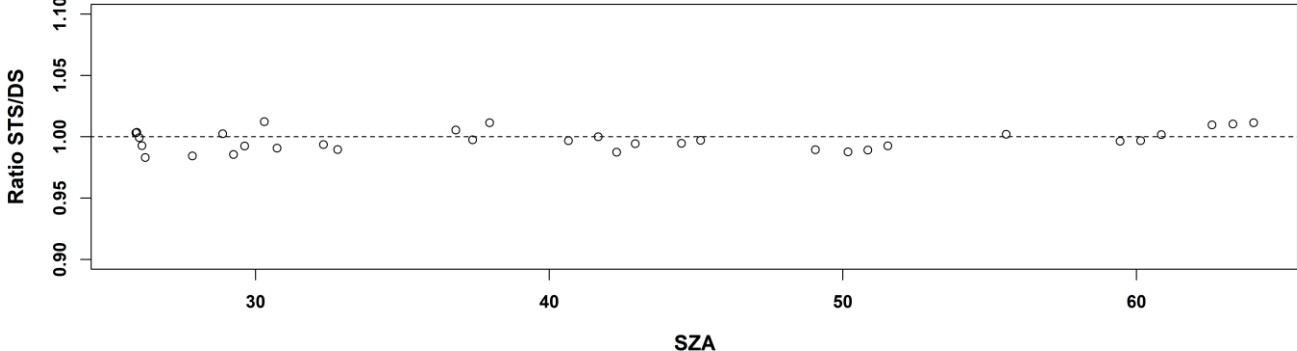

**Figure 8: The circles indicate the ratio between the STS TOC and DS measurements during day 49 from noon and on. The maximum deviation is 1.7 % and the standard deviation is 0.8 %.**

### 3.2 Radiative transfer modelling considerations

The whole concept of this method is the assumption that the model is able to reproduce the capabilities of the micro spectrometer within acceptable limits. In theory this means that all parameters that influence the measured irradiation in the

wavelength area of interest, can be reproduced by the model. Radiative transfer in the atmosphere and the interaction with land and sea is a highly complex phenomenon and practically impossible to fully reproduce through models. However, most of the parameters have approximately an equal effect on the irradiance with respect to the wavelength area of interest, within certain limits. In this case the wavelengths in use are around 312.2 and 321.9 nm where ozone has a much stronger absorption of UV radiation at 312.2 nm than 321.9 nm compared to other atmospheric constituents. By calculating the ratio

of irradiance between these two wavelengths, the change of the ratio is assumed to be equal for all parameters exept the TOC. As already mentioned, this assumption has clearly some limits which need to be accounted for. By assessing the data used in this work it is clear that clouds have the most negative effect on the results. Mostly the TOC results increase in value with more overcast conditions, mainly due to multiple scattering in the clouds (Mayer et al., 1998) but may also decrease in some cases when the clouds act in a radiative forcing manner (Estupiñán et al., 1996; Sabburg and Wong, 2000; Sabburg et

al., 2003; Schafer et al., 1996). In any case, the rejection of data in this study is based on how much the UV-irradiation is attenuated or enhanced, and a threshold of +/- 10 % irradiance at 340 nm from a clear sky standard atmosphere model was used. This means that all irradiance data with an attenuation or enhancement of 10 % or more are rejected in the analyses. The threshold value is chosen based on simple statistics with two criteria, where one is to obtain the lowest possible standard

deviation on the daily mean ratios between the STS and Brewer DS (fig 3, panel (b) ), but at the same time not losing more than 10 % of the number of days with measurements compared to the Brewer DS measurements.

As the specific aerosol profile, aerosol optical depth (AOD), albedo, and ozone profile are not known, they are a subject of uncertanty in the mesurements. For the calculation of the LUT, a rural type aerosol in the boundary layer and a background aerosol type above 2 km is assumed according to (Shettle, 1990). The albedo was set to 0.05 representing the area around the instrument site, visually observed as a mix of grassland and sand (Chadyšiene and And Girgždys, 2008; Mckenzie et al., 1996). The ozone profile was fixed to the standard mid latitude summer as described in (Anderson et al., 1986).

Assuming a TOC of 280 DU and and a change in AOD from 0.2 to 0.4, the modelled ozone changes by 7.4 DU (2.5 %). These AOD values are comparable to long term measurements in Rome (Campanelli et al., 2022), and could explain some of the variation in the STS measurements. Assuming an albedo of 0.1 reduces the TOC by 3.0 DU, and is an unrealistic assumption, and is probably negligible compared to the effect of clouds. Changing the ozone profile to the standard US profile (Anderson et al., 1986) gives an increase of ~3.0 DU. Since the STS measurements during the campaign does not have a significant trend, it's more likely that these factors mostly contribute to variations in the TOC estimate, rather than something systematic.

### 3.3 Date and time

For calculation of TOC through irradiance ratios, an accurate value of the SZA is important, and can be measured or calculated. For the STS the latter is used (Reda and Andreas, 2004), and accurate knowledge about date and time, and the instrument's geolocation is required. As the internal clock in the microcomputer running the STS may drift, it is important to keep track of the time as each measurement needs an accurate time stamp for the SZA calculation. The instrument was not allowed internet access (due to security issues) preventing accurate time keeping. Time was adjusted manually around every two weeks to keep an accuracy of +/- 20 sec to ensure the SZA calculation was within 0.1° accuracy, which is sufficient for the TOC calculation. With a SZA error of $\pm 0.1°$ at 60° for a TOC of 280 DU, the error estimate from the model calculation is only $\pm 0.7$ DU.

### 3.4 Instrument considerations

### 3.4.1 Temperature

Most detector based electronic instruments are temperature dependent in some way as higher temperature usually is related to more electrical noise. This problem is also evident in this case as the calculated STS TOC in general seems to increase with higher ambient temperature. Unfortunately, the STS instrument does not have a sensor for logging the instrument temperature. Instead, the Brewer reference instrument temperature log was used as the basis for temperature correction of the data. It was evident that for some days with extreme ambient temperatures (> 40° C) the STS could start to give erroneous data which could not be corrected for, and the data had to be rejected. The manufacturer claims a working temperature up to

50 °C, but it is likely to exceed this limit inside the instrument casing when it's exposed to sunshine in addition to extreme ambient temperatures. The only solution to collect data from these days was to include data from the morning or late afternoon only before it got too warm, assuming that data to be representative for the daily mean. Clearly, instrument temperature logging should be included for a more accurate correction. An even better solution would be a temperature regulation of the instrument which would require a more complex design and add to the cost, but it would most likely improve the performance as the temperature correction will no longer be needed. Simpler measures such as white painting, shading covers, and better ventilation could also be investigated as a way to reduce extreme temperatures at much lower cost.

### 3.4.2 Entrance optics

The performance of the instrument's entrance optics can be critical depending on the use, in particular when measurements of absolute values are of interest. When measuring irradiance, a cosine corrected collector is needed in order to correctly measure radiation from all downwelling directions. In the case of calculating TOC from irradiance measurements within the operating SZA area of the Brewer DS (< ~65°), ratios between irradiances are used and cosine correction errors are found to be relatively small and almost equal for the different wavelengths for different instrument types. Both the Brewer and the NILU-UV radiometer (Lakkala et al., 2018; Høiskar et al., 2003) have similar cosine response around 310nm – 320nm, using polytetrafluoroethylene (PTFE) as the diffusing material. The STS cosine collector was designed to be similar to the one used in the NILU-UV, and the STS results seem to prove the design to work for the purpose of measuring TOC.

### 3.4.2 Instrument maintenance

Measurements with optical equipment usually require frequent maintenance, in particular for outdoors measurements, where cleaning of the entrance optics is necessary. To see if absence of daily maintenance had any kind of impact, the STS instrument was maintained approximately only once a week. It is difficult to estimate the effect of less maintenance, but from inspection on days with rain showers it was clear that accumulation of water on the diffuser had a large impact until it evaporated or was dried off. No other effects like dust or bird droppings were observed, but it should be noted that the need for cleaning the entrance optics is highly site dependent. Urban areas usually have more aerosols which may deposit on the diffuser. In arctic and alpine areas snow showers often have a major impact, as the snow will cover the diffuser and could stay there over longer time periods until it blows or melts away. In these cases, frequent attention is required in order to reduce loss of data, however it is important to note that such work doesn't require a skilled technician.

### 4 Conclusions

The daily mean series of TOC from the STS instrument is comparable with both the Brewer DS and ZS series. However, it
was also found that the STS instrument with the current analysis software will significantly overestimate the TOC if the
global irradiance is reduced with more than 10 % in the 330 nm area due to clouds. The only short-term maintenance of the
instrument is removal of water/snow from the entrance optics after rain/snow showers, and any other deposits which may
accumulate. The stability over the six months period was very good and only interrupted by a power failure during a
thunderstorm. It was mostly possible to correct for the temperature dependency in the measurements for ambient
temperatures below ~40 °C. For improvement of the performance the temperature of the instrument could be regulated to a
fixed value. The instrument is very cost-effective, small, lightweight, and easy to operate.

As the spectrometer covers fully the UV-A/B spectrum it is fully possible to further develop the analyses software to include
erythemal UV-dose, vitamin D effective UV-dose, vitamin D effective UV-dose, UV-index, UV-albedo and wavelength
dependent cloud optical density.


*Author contributions.*

KE assembled the STS instrument, analysed the data and wrote the draft. SR provided the Brewer reference data, and both
MT and SR revised the manuscript.

We thank two anonymous referees for improvements to the manuscript.


*Competing interests.*

The contact author has declared that none of the authors has any competing interests.

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
