# Peer review of "Comparison of total ozone measurements in Melbourne, Australia, performed with a low-cost micro spectrometer and a Brewer MK-III"

_EGUsphere, 2025_

## Author Comment (AC1)

**We thank the reviewer for the thoughtful and constructive review.**

**RC1**

**Review of:**

**Comparison of total ozone measurements in Melbourne, Australia, performed with a low-cost micro spectrometer and a Brewer MK-III**

Kåre Edvardsen, Matt Tully, Steve Rhodes

Submitted to https://doi.org/10.5194/egusphere-2025-87 Preprint. Discussion started: 17 February 2025

**General comments:**

This paper provides a description of the methods to derive total column ozone (TCO) from a very low-cost instrument measuring global ultra-violet irradiance. The paper is well written, well structured and the methods are clearly described. It is recommended for publication after addressing the comments below.

The paper focuses rather on the method of TCO retrial rather than a sound analysis of the performance of TCO compared to a reference instrument. Therefore, the general comments are divided into two following parts:

Method:

Commonly, TCO is derived best by direct sun measurements. The paper uses global UV measurements, which implicate parameters of uncertainty such as clouds in front of the sun, clouds in the sky, effective albedo from the surrounding terrain and cosine response of the diffusor. All these effects would not occur when using direct sun measurements. The authors argue that in many places worldwide cloud coverage hinder direct sun measurement for TCO. Are there so many places with such conditions? Maybe the performance of the low-cost instrument would be much better when using direct sun measurements. Has this ever been tested with the instrument?

**This is a fair question but would have to be the subject of future work. The GI configuration was chosen in part to keep the design as simple as possible, and to assess what performance could be achieved with the least complexity. Many locations around the world experience very persistent cloud cover in certain seasons or synoptic conditions, so in principle an instrument that could only measure ozone in sunny conditions will lead to bias in the ozone record.**

The method includes many parametrizations based on empirical functions. I have lost the overview. Maybe the authors can provide a summary of all the empirical parametrizations, the number of measurements needed for the parametrization and, if possible, an estimation of the uncertainty of the parametrization and its implication of TCO.

The parametrization also arises the question if the method is also suitable for other placed world wide, once it has "calibrated" with a Brewer at a specific site. Or must the method be "calibrated" for each location worldwide separately with a Brewer?

**The method used on the micro spectrometer data (STS GI) is not based on parametrizations of empirical functions, like the Brewer "zenith sky" method. TOC retrieval from GI measurements is based on the multi-stream discrete ordinates radiative transfer equation solver (see line 135 and on). The model (LibRadtran) simulates the instrument's response to the global irradiation $I_{321.9\,nm}$ and $I_{312.2\,nm}$ and the TOC is derived from when the measured ratio $R_{me} = \frac{I_{321.9\,nm}}{I_{312.2\,nm}}$ fits the model ratio $R_{mod}$ for the same global irradiance ratio with TOC as the model input variable.**
**The method does not need a calibration for each new location once the instrument is "calibrated" with a Brewer. However, this requires that other significant UV sensitive parameters are accounted for (albedo, aerosols etc). For the wavelengths used, other UV sensitive parameters than ozone, have an almost equal effect on the two wavelengths, so they cancel out in the ratio.**

**The results reported here are based on only one location.**

**However, the very same method is in use by the Norwegian UV and ozone measurement network. This network consists of two Brewers and eight Biospherical GUV instruments measuring ozone from Olso (60 N) to Ny-Alesund (78 N) with similar results (line 80 and on).**

Performance:

It is not clear if the authors have compared daily averages with quasi synchronous measurements for STS GI and Brewer DS. I suggest comparing synchronous measurements within e.g. 10 minutes intervall and also show the individual measurements to indicate the variation of TCO.

**Daily means are based on all available data for each instrument within the Solar Zenith Angle limits the Brewer was measuring TOC within, so there is no filtering for quasi synchronous measurements. The micro spectrometer recorded data continuously with a time resolution of one minute, and synchronous measurements are shown in figure 4 (Brewer data is red, STS data in black and green), but this is only for one day where the TOC seem stable. The TOC does not vary very much at this latitude (less than 10 DU 95 % of the days), but we have added figure 6 that indicates the variation of the TOC when the variation was 22 DU, and some comments (line 270-275 in the revised manuscript.)**

Commonly these small low cost spectrometers suffer from straylight, which biases TCO at low solar zenith angles or high ozone slant columns. In order to assess the performance of the instrument the comparison between the Brewer and STS GI depending on ozone slant column (airmass * TCO) should be provided.

**Yes, the method certainly breaks down at high solar zenith angles (we are sure the reviewer means high solar zenith angles rather than low). We have added figure 7 which shows this clearly. This issue was not discussed in detail because the Brewer's schedule during the**

campaign did not include any TOC measurements with higher zenith angles than ~65 degrees, as the Brewer making Umkehr measurements at these times.

Temperature dependency. Indeed, the dependency of the small array spectroradiometer is a crucial issue of these instruments. Ambient temperature causes wavelength shifts, linearity and detectability problems (signal to noise ratio etc.). I am surprised that a simple parametrization (Eq. 2) can account for that. It would be worthful to calculate the uncertainty of TCO based on the parametrization and its applicability for other instruments and other locations.

**We did not attempt to conduct a full uncertainty analysis for this work.**

**A very useful property of this particular spectrometer is that the detector is sensitive all the way down to around 190nm, and the noise in the detector can be established from the blind area of 180 – 280 nm continuously. Then the only major thing left is to find the temperature effect on the TOC calculations, which turned out to be quite simple in this case. The temperature correction is further elaborated well described in section 2.5 with added plots (page 8)**

Furthermore, the model includes the ozone absorption cross section, which depends on the effective ozone temperature (basically stratospheric temperature). How sensitive is the algorithm and resulting TCO on effective ozone temperature?

**We did not attempt to calculate the sensitivity of the results to effective stratospheric ozone temperature. This is a relatively small effect at mid-latitudes even for reference ozone instruments (Voglmeier et al. ). For the very simple, low cost instruments being studied, we have focused on the major issues such as clouds and ambient temperature. We agree though that in in principle a small seasonal and latitudinal error would be expected from this cause.**

Specific comments:

Page 2: line 32: How significant is the fraction of the price of the Brewer. To my estimate it is about 10% of a Brewer, which is already low cost.

**Prices vary of course with time, delivery costs, currency conversions etc, but in broad terms an instruments such as the Pandora or BTS would cost at least 20-30% of the price of a Brewer. This instrument is more than an order of magnitude cheaper so we feel the statement is quite reasonable.**

Page 2, line 47: Is the MK II Brewer a double monochromator with insignificant stray light impact?

**The Brewer was Mk III with a double monochromator and low stray light.**

Page 2, line 60: Can the method be described as a cloud correction?

**No, the alternating DS and ZS measurements establish a pure statistical relationship by solving for the nine coefficients (a -k) in equation (1) regardless of clouds.**

Page 3 line 89: The reference is missing

**Thanks. It has been added in the revised version.**

Page 4 line 95: What is the slit function and full width half maximum of the instrument, resulting from 25 micron slit?

**The manufacturer claimed 4 pixels FWHM (~1.5 nm) slit function from a lab test (Hg 253.6 line), but the instrument simulations from the LibRadtran model predicted a complete instrument function of ~3.0 nm FWHM at 312 nm, which was used in the calculations. This information has been added to the manuscript.**

Page 5 line 126: Do you mean ozone slant column (=airmass * TCO) - > see comment above.

**No, ozone slant "path" was intended – ($\mu$) refers to the path length through the atmosphere, relative to the vertical path length (often referred to as "air mass factor" in the Brewer community).**

Page 5: 132: and EQ 1 is this parametrization also applicable at other locations

**Equation (1) is for the Brewer ZS measurements used in absence of Brewer DS measurements, and should be calculated for each Brewer at each site.**

Page 5, line 146. Again, what is the full width half maximum of the slit function?

**It is described according to the question above.**

Table 1: It would be worthful to indicate the performance in percent

**The table now includes the percentage.**

Caption Figure 2: SZ -> ZS

**It is corrected.**

Figure 4. The ratio between Brewer and STS would be more helpful

**A ratio plot for the same data as fig 4 is presented in fig 8.**

Page 11 line 280: Why is TCO increasing with temperature? Wavelength shift?

**Yes, we would expect wavelength shift, which is in general related to instrument temperature, to be the major cause.**

Page 11 line 286: I suggest using quasi simultaneous measurements.

**Thanks for the suggestion. We have chosen instead to add a new figure 2 (panel a) clearly demonstrating the temperature effect on the measurements, and how well the correction works after being applied (panel b and c). We hope this addresses the concern.**

Page 12 line 310: At what conditions is STS comparable to the Brewer? What are the limitations?

**As mentioned already in the abstract, clouds and temperature is the main contributor to the rejection of measurements. The cloud limitation is outlined in section 2.4 lines 181 – 192 in the revised manuscript, and the temperature limitation in section 3.4.1 lines 349 – 362.**

Page 13: line 324. The conclusion should not end with a bullet list. Maybe a closing sentence would help.

**Thanks, yes, the bullet list has now been replaced with a closing sentence.**

---

## Author Comment (AC2)

**We thank the reviewer for the very thorough, detailed and constructive review. The reviewer makes a number of very pertinent suggestions for improvements to both the instrument and the retrieval which we hope to investigate with future work and further development. Of course, the results reported here relate to a specific campaign carried out over a fixed time period which was not able to be extended.**

**RC2**

**General Comments:**

The paper by Edvardsen et al. presents a newly developed low-cost total ozone instrument, including technical descriptions, ozone retrieval algorithm, and some validation. The new instrument shows good potential to be a low-cost instrument performing network-level monitoring. However, the retrieval algorithm currently developed seems to be oversimplified and some extra validation work would be good to confirm its performance. In addition, the calibration of the new instrument relies on a collocated Brewer spectrophotometer. Is the calibration at a given site transferable to any other locations (where we might not have a Brewer)? Most likely it will not be straightforward, such as it would depend on the altitude of the site, local ozone and aerosol profiles, etc. Some discussion and explanation is needed.

Nevertheless, this work is a good match with AMT and within the scope of the journal. I recommend publishing this work after addressing the following issues.

**While the work reported here is based on the results of a specific comparison at one location, it should apply also at other mid-latitude sites.**

**Specific Comments:**

Line 32: to make this claim more convincing, maybe give some rough numbers here for the listed instruments. At least, Pandora and BTS-solar instruments are commercially available.

**We would prefer not to state numbers because prices vary of course with time, delivery costs, currency conversions etc, but in broad terms an instruments such as the Pandora or BTS would cost at least 20-30% of the price of a Brewer. This instrument is more than an order of magnitude cheaper than that so we feel the statement is quite reasonable. As the reviewer intimates, Brewers are not currently available for purchase in any case.**

Line 55-56: In fact, there are still lots of good improvements on Brewer DS ozone. E.g., Savastiouk et al., 2023.

**Thanks, this has been added.**

Line 57-60: To justify the claim, some more quantitative description is needed. E.g., % of DS data removed due to cloud conditions.

**Thanks, this has been added.**

Line 60-64: some references for Brewer ZS method and data quality are needed.

**Reference to the Brewer ZS method is found at the end of section 3.1.**

Line 84-85: The cloud effect could affect DS, ZS, and even GI TOC. I understand the latter two should have better performance than DS in cloudy conditions. Some more quantitative description is needed to support such a claim.

**We have added additional quantitative description in section 3.1 Data assessment.**

Line 110: Why does Brewer DS have a cut off at 63 degrees (mu value around 2.1)? Is this a single or double Brewer? For double, typically the cut off could be 75 degrees (mu around 3.5).

**It is a double Brewer but at the time of the campaign, the schedule had been set to only take Umkehr measurements above 63° . The information is added to the manuscript.**

Line 116: is there an option to control the exposure time? It is a bit surprising that some data has to be discarded due to this.

**Yes, the exposure time can be set to a fixed value in the operating software up to a maximum of 10s. It was initially set too high at the beginning of the campaign which caused the detector to saturate for wavelengths > 310 nm, for SZA's less than ~40° - 45°. Once the problem had been identified a more optimal setting was used.**

Line 120-121: some observation SZA ranges could be provided for the reader to understand the capacity of STS. Can STS perform observations when SZA is close to 90 degrees (like ZSL-DOAS systems)?

**Figure 6 shows how the measurements are unreliable for SZA's > ~70°. From inspection of the raw data the dark level at 312.2 nm is ~80% of the total signal when the SZA reaches ~70° resulting in a very low dynamic range (see last paragraph of 3.1).**

Line 165: given this is a new effort, why use "Bass and Paur", not anything we know has better performance (e.g., SDY, see Redondas et al., 2014; Voglmeier et al., 2024)?

**Bass and Paur was chosen to match the cross-sections used by the Brewer algorithm.**

Line 155-156: The figure shows the method will lose sensitivity for SZA< 40 degrees. The good news is the method would have good sensitivity for TOC from 300-500 DU, when SZA is in the range of 40 to 80 degrees. This should be good enough for most mid-latitude conditions (but could be challenging for low- and high- latitude areas). I would expect to see a detailed exam on the algorithm detection limits or uncertainty budgets.

**We don't agree not that the method loses sensitivity for SZA < 40° as the detector is working in the higher level of the dynamic range (0-16384) and the UV-irradiance is much stronger compared to SZA's > 40°. Towards SZA of 70° the dynamic range is reduced 5-6 times compared to SZA's ~ 30° (see last paragraph of 3.1), and this is of course a problem. For all realistic TOC and SZA combinations for any latitude, the best results are always obtained**

**for smaller SZA's. The problem of high latitude areas (SZA's > 40° / large air mass) is in general weaker UV-irradiance in combination with generally cloudier conditions and higher TOC values in these areas, further reducing the UV-irradiance.**

Line 161-162: So, the LUT is only good for mid-latitude, correct?

**Yes, libRadtran offers the use of AFGL atmosphere profiles from tropical, mid-latitude, and sub-arctic areas. The LUT for this work was calculated using a mid-latitude profile. Further work could consider the sensitivity of the results to the details of the profile used to calculate the LUT.**

Line 172-175: I am a bit surprised that the author directly used the ratio between measured and modelled irradiance. Given that the spectrometer is not absolute calibrated, how can we ensure such ratio close to unity? Such ratio will be different for different instruments, correct?

**We have revised the text to try to make the explanation more complete and hopefully less confusing. The spectrometer was quasi calibrated towards a clear sky situation at 340 nm using a modelled GI$_{340nm}$ as the reference**

Line 193-196: I am concerned about this issue. First, this 5% offset per 10 degrees C is not small. For some days in mid-latitude regions, a 10-degree C daily variation is common. The temperature for the warmest and coldest days is not given. But, if we use the 44.3 degrees C from the second warmest day (reported in the paper), the correction (or bias) is alarmingly 12%. Is this from wavelength drifting? For the algorithm, is there any "dynamic" wavelength registration or is it just using values from fixed pixels? Is this STS system placed outdoors without any temperature control? In any case, the author should show the correction factor's linear regression plot here (and also linear regressions of STS TOC vs. Brewer TOC, before and after applying this correction). Sophisticated correction is "cheaper", only if we cannot do it correctly. I agree with the author that some level of T control is needed for this system.

**This is a fair point. In the configuration used in the campaign, there was no way to know the temperature inside the STS, so the Brewer temperature sensor was used as a proxy. The analysis has used values from the same range of pixels for the two wavelengths regardless of temperature (that is, no dynamic wavelength registration). The instrument was left outdoors without any temperature control.**

**The temperature correction factor's linear regression and the linear regressions of STS TOC vs. Brewer, before and after applying this correction is shown in figure 2 in the manuscript.**

**We believe we have been transparent and accurately represented the performance and the limitations of the instrument as implemented in this campaign.**

Line 207-208: why STS has less data points than Brewer DS and ZS? Even including those four days of thunderstorm, STS still has fewer data points than others. Any idea? I would expect that this STS GI method should produce more data even in cloudy conditions, based on the claims in previous sections.

**In addition to the thunderstorm there were some other operational issues which reduced the number of measurements. One was the saturation problem caused by a too-high setting of the integration time which wasn't immediately noticed. Water collecting on the diffuser after rainfall also effected some days.**

Line 238-251: it is a bit strange to show a trend for such a small dataset, which even could not cover a full season cycle. But, it is a bit surprising that Brewer DS and ZS ratio has a significant trend. If this trend is real and keeps increasing, some ZS calibration might be needed (i.e., the nine coefficients should be recalculated). When was the last DS and ZS calibration done for this Brewer? Could the author include scatter plots for such a comparison? Also, I agree with the author that the poor performance of STS in the first 30 days could be due to inaccurate T correction. I am wondering if the author calculated the T correction parameters based on daily mean data or high-resolution observations. Any estimation on the uncertainty from this correction?

**Yes, the trend is statistically significant, but the effect is in only in the order of + 0.9 DU per month on average during the campaign. Since the ZS coefficients can be recalculated, the ZS results probably can be improved. However, the authors think that a scatter plot comparison to the ZS is a bit off the scope of this work as the STS measurements are mainly compared to the standard Brewer DS measurements.**
**Not only could the poor performance the first 30 days be due to inaccurate T correction, but also to the fact of the detector saturation for SZA < ~45 degrees. On average, 4 hours of data around solar noon during January is disregarded.**
**The temperature correction was applied to the high resolution data before further calculations were made. To put a number on the uncertainty from the temperature correction is really challenging as it is impossible to know the accurate STS temperature at any time. However, the difference in the standard deviation of the residuals between the uncorrected and corrected daily mean (Fig 2, panel b and c) at least gives an estimate on the uncertainty. Last DS calibration was May 22, 2018 and the ZS coefficients were recalculated March 21, 2019. The information is provided in the manuscript, line 99 – 100.**

Line 261-267: Not just cloud but also aerosol and surface albedo could affect the results. Cloud optical depth, aerosol optical depth, cloud layer heights, aerosol layer heights, surface albedo, and ozone profiles could all affect the ratio. This simple ratio method reminds me of "color-index" commonly used in the DOAS community, which is also just a simple ratio of two wavelengths (e.g., Wagner et al., 2016; Zhao et al., 2019). By any means, to convince the reader that all the suggested factors won't show a big impact, some detailed modelling work should be done (even better to understand the uncertainty).

**Modelling work on aerosol optical depth, albedo, and ozone profile shows less impact on the results. The modeling results are added at the end of section 3.2.**

Line 275-276: A typical timeserver nowadays can assure more than enough accuracy than ±20 s. The major issue for this method and instrument came from other places. This is why I would suggest some level of uncertainty in budget estimation.

**The instrument was not allowed network access at the site (security issues), so time had to be adjusted manually around every two weeks to assure less than a 20 s drift. Maximum SZA change during any day is less than 0.2 deg/min. With an error of 0.1 deg @ SZA = 60 deg, the error is only 0.7 DU @ TOC = 280 DU. The error estimate is added to the manuscript.**

Line 279-281: based on the information provided, this is not a simple electrical noise, but something systematic. My first guess would be the wavelength registration issue. Is there any compensation for wavelength drifting? The resolution of the selected spectrometer is pretty low (3 nm), but do we see any slit function changes?

**We agree that there appears to be some systematic effect here which is probably related to wavelength shift. Our investigations were not able to characterise the drift in any way though which produced better results than the simple linear correction based on the proxy (Brewer) temperature.**

Line 287-289: Simple solutions such as white painting, shading covers, and better ventilation could be done to reduce direct heating from the sun. In any case, if the system could only survive up to 40 degrees C, it could not be used in many places. Also, is the simple linear correction for T only valid within 40 °C? Note that in Line 199, it claims that the system could work from 0 to 50 °C. Please provide consistent information.

**Yes, these are all very useful suggestions which definitely would have reduced the risk of overheating when outdoors temperature exceeded 40 °C. For a subsequent campaign we would definitely pay more attention to both temperature regulation and temperature characterisation.**

**The claim in line 199 is not for the system overall, but just for the spectrometer itself as it is not specified for operation in environments above 50 °C. Consistent information is provided in the revised manuscript.**

Line 310-312: some quantitative description is needed. It is a bit hard to understand to what extent STS consider the condition as "too cloudy".

**"Too cloudy" is replaced with a more quantitative explanation in the revised manuscript.**

Line 313-314: give this is a very simple instrument, such stability for six months is good but not enough. Lots of total ozone monitoring instruments need to be as stable as 2 years with minimal maintenance supports (such as entrance cleaning). Also, it would be important to see if there is any seasonal bias between STS and Brewer. Is this STS no longer co-located with Brewer?

**Yes, this is all very true, but the campaign was limited by circumstances and not able to be extended. In the future we hope to repeat the comparison for a longer period of time but with some refinements.**

Last, in fact, Brewer also can retrieve total ozone via UV irradiance, not just DS and ZS. Some solid works were done almost 30 years ago. I would suggest the author check (Fioletov et al., 1997). For example, I would suggest using the log scale for the ratio and using all wavelengths instead of just one (312 nm).

**Thanks, we have added the reference and noted this in the text.**

**We found through lab testing that other wavelengths in this range (305-323 nm) suffered from non-linear response and concluded that the only pair giving reasonable results was the 312/322 nm pair. This might have been a characteristic of the individual spectrometer though.**

**Technical Comments:**

Line 26: left parenthesis is missing

**OK**

Line 82-84: need to rewrite this sentence to make it clearer

**OK**

Line 89: reference is missing

**OK**

**Line 91: definitions for WOUDC, NDACC, and PGN are needed.**

**OK**

Line 93: change "Environment & Climate Change, Canada" to "Environment and Climate Change Canada".

**OK**

Line 102: delete the repeated "in a"

**OK**

Line 112: change "clods" to "clouds"

**OK**

Figure 2: what are "SZ" measurements? Please use consistent naming for the data. Also, the legend shows DS, ZS, and STS, while the caption talked about "ZS and GI measurements".

 **OK**

**Reference**

Fioletov, V. E., Kerr, J. B., and Wardle, D. I.: The relationship between total ozone and spectral UV irradiance from Brewer observations and its use for derivation of total ozone from UV measurements, Geophys. Res. Lett., 24, 2997–3000, https://doi.org/10.1029/97GL53153, 1997.

Redondas, A., Evans, R., Stuebi, R., Köhler, U., and Weber, M.: Evaluation of the use of five laboratory-determined ozone absorption cross sections in Brewer and Dobson retrieval algorithms, Atmos. Chem. Phys., 14, 1635–1648, https://doi.org/10.5194/acp-14-1635-2014, 2014.

Savastiouk, V., Diémoz, H., and McElroy, C. T.: A physically based correction for stray light in Brewer spectrophotometer data analysis, Atmospheric Measurement Techniques, 16, 4785–4806, https://doi.org/10.5194/amt-16-4785-2023, 2023.

Voglmeier, K., Velazco, V. A., Egli, L., Gröbner, J., Redondas, A., and Steinbrecht, W.: The transition to new ozone absorption cross sections for Dobson and Brewer total ozone measurements, Atmospheric Measurement Techniques, 17, 2277–2294, https://doi.org/10.5194/amt-17-2277-2024, 2024.

Wagner, T., Beirle, S., Remmers, J., Shaiganfar, R., and Wang, Y.: Absolute calibration of the colour index and $O_4$ absorption derived from Multi AXis (MAX-)DOAS measurements and their application to a standardised cloud classification algorithm, Atmos. Meas. Tech., 9, 4803–4823, https://doi.org/10.5194/amt-9-4803-2016, 2016.

Zhao, X., Bognar, K., Fioletov, V., Pazmino, A., Goutail, F., Millán, L., Manney, G., Adams, C., and Strong, K.: Assessing the impact of clouds on ground-based UV–visible total column ozone measurements in the high Arctic, Atmos. Meas. Tech., 12, 2463–2483, https://doi.org/10.5194/amt-12-2463-2019, 2019.